

# Signature identification of relapse-related overall survival of early lung adenocarcinoma after radical surgery

Peng Han, Jiaqi Yue, Kangle Kong, Shan Hu, Peng Cao, Yu Deng, Fan Li and Bo Zhao

Department of Thoracic Surgery, Tongji Hospital, Tongji Medical College, Huazhong University of Science and Technology, Wuhan, Hubei, China

## ABSTRACT

**Background**. The widespread use of low-dose chest CT screening has improved the detection of early lung adenocarcinoma. Radical surgery is the best treatment strategy for patients with early lung adenocarcinoma; however, some patients present with postoperative recurrence and poor prognosis. Through this study, we hope to establish a model that can identify patients that are prone to recurrence and have poor prognosis after surgery for early lung adenocarcinoma.

**Materials and Methods**. We screened prognostic and relapse-related genes using The Cancer Genome Atlas (TCGA) database and the GSE50081 dataset from the Gene Expression Omnibus (GEO) database. The GSE30219 dataset was used to further screen target genes and construct a risk prognosis signature. Time-dependent ROC analysis, calibration degree analysis, and DCA were used to evaluate the reliability of the model. We validated the TCGA dataset, GSE50081, and GSE30219 internally. External validation was conducted in the GSE31210 dataset.

**Results**. A novel four-gene signature (INPP5B, FOSL2, CDCA3, RASAL2) was established to predict relapse-related survival outcomes in patients with early lung adenocarcinoma after surgery. The discovery of these genes may reveal the molecular mechanism of recurrence and poor prognosis of early lung adenocarcinoma. In addition, ROC analysis, calibration analysis and DCA were used to verify the genetic signature internally and externally. Our results showed that our gene signature had a good predictive ability for recurrence and prognosis.

**Conclusions**. We established a four-gene signature and predictive model to predict the recurrence and corresponding survival rates in patients with early lung adenocarcinoma after surgery. These may be helpful for reforumulating post-operative consolidation treatment strategies.

## INTRODUCTION

According to the latest epidemiological studies, lung cancer remains the most common malignancy in the world and one of the leading causes of cancer-related death (*Siegel et al., 2021*). It can be divided into small cell lung cancer (SCLC) and non-small cell lung cancer

Corresponding authors
Fan Li, tjhtsdrli@163.com
Bo Zhao, 13006369600@163.com

(NSCLC). NSCLC accounts for approximately 85% of all lung cancer cases. NSCLC can be categorized as lung adenocarcinoma (LUAD) and squamous cell lung carcinoma. according to its pathogenesis and histological morphology (*Molina et al., 2008*; *Malhotra et al., 2016*). As tobacco control increases, lung adenocarcinoma incidence is now higher than that of lung squamous cell carcinoma and has become the main cause of lung cancer-related death (*Siegel et al., 2021*; *Cayuela et al., 2021*). However, lung cancer prognosis is still poor. There are no characteristic clinical symptoms in the early stages of lung cancer, delaying its diagnosis and treatment (*Jacobsen et al., 2017*; *Hong et al., 2015*). We rely on low-dose chest CT scan more than clinical symptoms, chest X-rays, or tumor markers for the early diagnosis of lung cancer (*Cainap et al., 2020*; *Zhang et al., 2019a*). Therefore, the application of low-dose chest CT scans in the screening of early lung disease is becoming more common. It is worth noting that, with the outbreak of COVID-19, low-dose chest CT scanning was widely popularized in the early screening of the disease (*Herpe et al., 2021*), which indirectly promoted the early diagnosis of lung cancer (*Zhang et al., 2020*). The diagnosis and treatment of lung cancer, including surgical resection, chemotherapy, radiotherapy, targeted therapy, and immunotherapy, have improved (*Yuan et al., 2019*; *Wang et al., 2020*; *Zappa & Mousa, 2016*). Among all the treatment methods, radical surgical resection is the first choice for treating patients clinically diagnosed with early lung cancer and can significantly improve their prognosis (*Abbas, 2018*). Over the past decade, the widespread popularization of thoracoscopy in thoracic surgery has reduced the surgical trauma for patients and has further established the therapeutic status of radical surgery (*Abbas, 2018*; *Leshnower et al., 2010*; *Shaw et al., 2008*). Ideally, we hope to detect lung cancer early through regular physical examination and then carry out radical minimally invasive surgical treatment to completely cure the cancer. However, some patients are prone to relapse after radical surgery, which seriously affects the prognosis and patients' quality of life (*Yang et al., 2020*). Targeted therapy and immunotherapy have dramatically improved with remarkable effects in recent years; however, there is still no conclusion as to whether consolidation therapy such as targeted therapy or postoperative immunotherapy is necessary to improve disease outcomes. We do not have a comprehensive and systematic method to identify this disease, which limits postoperative survival for patients with early-stage lung cancer.

Gene chip and high-throughput sequencing technology are becoming more popular. We analyzed RNASeq datasets from The Cancer Genome Atlas (TCGA) database and gene-chip datasets from three GEO datasets. Univariate Cox regression analysis and Lasso regression analysis were used to identify common genes associated with prognosis and recurrence. A model predicting the relapse-related overall survival rate of early lung adenocarcinoma was established and validated. In contrast to previous reports on lung adenocarcinoma prognosis and recurrence models (*Merritt et al., 2020*; *Jones et al., 2021*; *Li et al., 2017*; *Yin et al., 2020*), our model was dedicated to predicting both overall survival and relapse-free survival in patients with early postoperative lung adenocarcinoma. Our model contains two prognostic factors which may be more clinically applicable. Evaluating the survival benefit based only on postoperative recurrence does not provide a comprehensive view. A subset of exceptional recurrent patients may present with slow-growing recurrent

pulmonary nodules (*Kobayashi et al., 2015*; *Detterbeck, 2019*; *Kobayashi & Mitsudomi, 2019*) and survival with nodules rather than overtreatment may be a better clinical strategy (*Hammer & Hatabu, 2020*). We defined early lung cancer as T1 and T2 patients regardless of N stage. Ours conclusion is prospective and applicable to clinical scenarios. In clinical practice it is easy to confirm the T stage of patients by chest CT scan, but it is difficult to confirm the N stage of patients (*McPherson et al., 2020*; *Ghaly et al., 2017*). Most patients will choose to undergo radical surgical treatment as early as possible in the T1/T2 stage of tumors.

## MATERIALS AND METHODS

### Data processing

We downloaded RNA-sequencing datasets for 551 LUAD tissue samples with corresponding clinical data from TCGA (https://portal.gdc.cancer.gov/) on November 22, 2020. Exclusion criteria included: (1) a history of prior malignancy or other existing malignancy; (2) patients who received neoadjuvant therapy or other unacceptable treatment; (3) unclear pathological diagnosis or adenocarcinoma with mixed subtypes; and (4) patients with T stage III/IV, distant metastases, or uncertain TNM staging. We included 239 LUAD samples in a preliminary univariate survival analysis. Batch effect removal and differential expression analysis were performed using the DEseq2 package (http://www.bioconductor.org/packages/release/bioc/html/DESeq2.html). The prognostic related genes were identified based on an adjusted $P$ value < 0.05.

Three Affymetrix datasets (GSE50081, GSE30219 and GSE31210) from the GEO database were screened out and included in our next analysis to ensure the reliability of our results. The screening conditions included: (1) samples in the dataset had clear pathological diagnosis, TNM staging, and prognosis information; (2) early LUAD samples of >50 cases; (3) postoperative sample availability; and (4) overall survival time and recurrence time was greater than 30 days. We used R (version 4.0.2) for our analysis. The Robust multi-chip Averaging (RMA) algorithm of the "Affy (*Gautier et al., 2004*)" package directly extracted data for processing.

### GO and KEGG pathway enrichment analysis

Gene Ontology (GO) and Kyoto Encyclopedia of Genes and Genomes (KEGG) pathway enrichment analysis was performed to comprehensively evaluate the biological processes of prognostic differentially genes to explore the potential molecular mechanisms behind prognostic genes. GO enrichment analysis was conducted to analyze the biological functions of prognostic-related differentially genes of early LUAD from cellular component (CC), molecular function (MF), and biological process (BP). KEGG enrichment analysis was dedicated to predicting the major biological pathways involved in DEGs. An adjusted $P$ value < 0.05 was considered statistically significant. We used the org.hs.eg.db and clusterProfiler (*Yu et al., 2012*) R packages for enrichment analysis.

### Establishing and validating a prognostic predictive signature

We performed prognostic analysis on the TCGA datasets, the GSE50081 survival dataset, and the GSE50081 relapse dataset. The intersection was taken to obtain relapse-related

prognostic genes. Subsequently, the above genes were included into the GSE30219 dataset for univariate regression analysis and Lasso regression analysis. The selected genes were used to construct a multivariate Cox proportional risk regression signature, and the corresponding risk score was calculated to evaluate the prognosis and recurrence of the patients. Risk score = $\beta1$*Exp1+ $\beta2$*Exp2+ $\beta3$*Exp3..., where Exp is the normalized gene expression level and $\beta$ is the coefficient value. According to the normality test, the appropriate cut-off value was calculated to divide LUAD patients into low- and high-groups. The Kaplan–Meier method was used to evaluate the difference in overall survival and recurrence between low- and high-groups. The K-M analysis was conducted to validate our risk signature. The R packages used in this analysis include "Survival", "plyr", "ggplot2", "SurvMiner", "dplyr", "Formula", "tidyverse", and "Survivalroc".

## Constructing and evaluating a predictive nomogram

We calculated the survival and relapse related clinical parameters (gender, age, smoking, T stage, N stage, and risk score) through univariate and multivariate regression analyses to improve the clinical transformation efficiency of our model. We used filtered-independent prognostic clinical parameters and risk scores to construct a univariate nomogram and a combined nomogram. Three recognized algorithms were applied for internal and external validation, respectively, to confirm the predictive potential of those models. ROC analysis was applied to evaluate the predictive performance of the nomogram. Calibration analysis was selected to evaluate clinical consistency of the histogram predictions. We used DCA to evaluate the clinical net benefit. For internal validation, the above three validation methods were used to evaluate one-year, three-year, and five-year OS and RFS of early LUAD patients in the TCGA, GSE50081, and GSE30219 datasets, respectively. The same method was repeated in the GSE31210 dataset for external evaluation. The R packages applied in this study included "ROCR30", "SurvMisc", "RMS", "pROC", "hash", "timeROC", "forestplot" and "VennDiagram".

## RESULTS

### Preliminary identification of relapse-related prognostic genes

We thoroughly screened the LUAD RNAseq data downloaded from the TCGA database and included 239 tumor samples in our prognostic analysis. The flow chart is shown in Fig. 1. We removed 12 patients with an OS <30 days or with unknown survival information and the remaining 227 patients were included in our univariate Cox regression analysis. The results indicated that 1,455 protein coding genes ($P < 0.05$) were statistically significant, including 567 differentially up-regulated genes and 410 down-regulated genes (239 Tumor *vs.* 36 Normal, $P < 0.05$) (Table S1). We then screened the GSE50081 dataset, which had complete TNM staging, prognosis, and relapse-related information, to further identify statistically significant relapse-associated prognostic genes. After normalizing using the RMA algorithm (Fig. 2A), univariate regression analysis was performed. Genes related to survival prognosis (Table S2) and disease-free survival (Table S3) were calculated, respectively. We recognized 948 genes associated with LUAD survival prognosis ($P < 0.01$) and 2,698 genes with LUAD recurrence ($P < 0.05$). We intersected GSE50081 survival

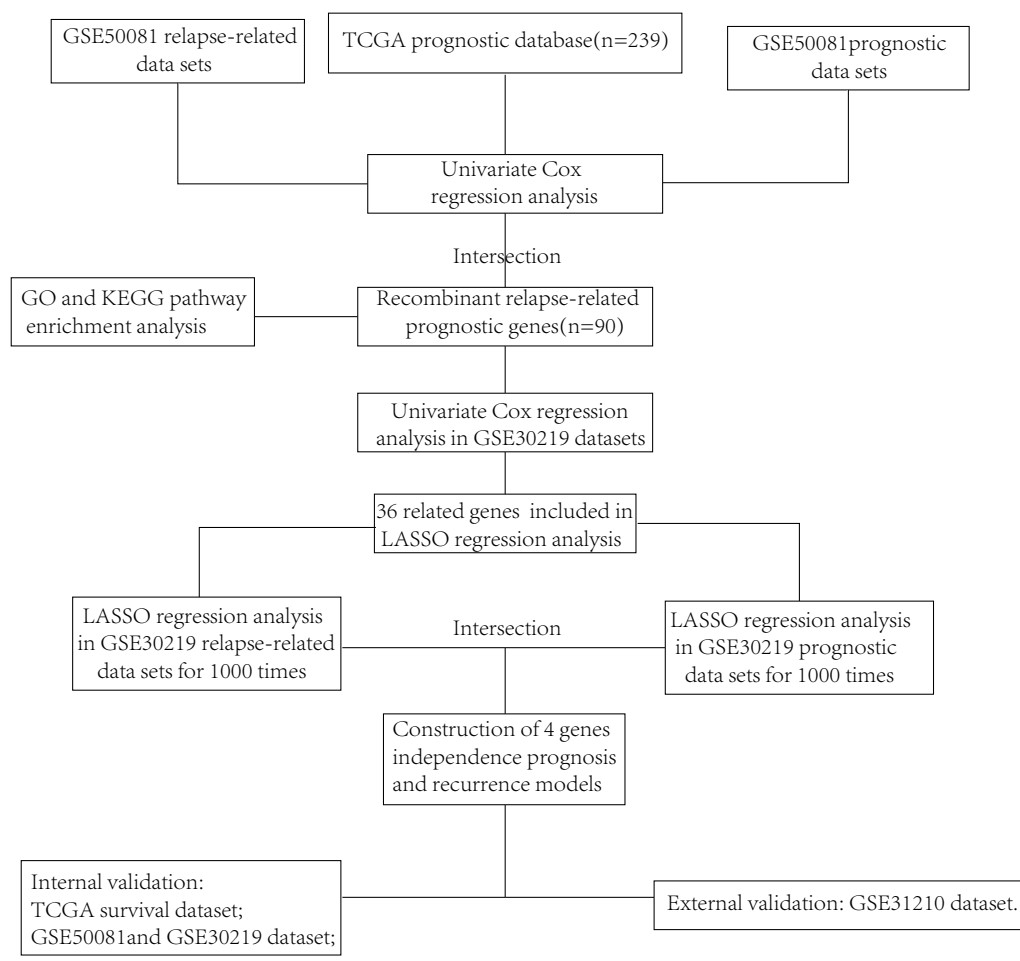

**Figure 1** **The flow chart showing the scheme of our study on relapse-related prognostic signatures of early LUAD.**

related genes, GSE50081 disease-free survival related genes, and TCGA prognostic related genes. We then obtained an additional 90 reliable relapse-related prognostic genes (Fig. 2B).

## GO and KEGG pathway enrichment analysis

GO enrichment analysis was conducted on 90 relapse-associated prognostic genes to explore the major biological functional modules and pathways involved in prognostic genes. Our results showed that those genes were mainly enriched in biological processes related to "platelet-derived growth factor binding", "extracellular matrix structural constituent conferring tensile strength", "growth factor binding", "proteoglycan binding", "aminopeptidase activity", "cyclin-dependent protein serine/threonine kinase regulator activity", "NADP binding", "cell adhesion molecule binding", and "coenzyme binding" (Fig. 2C). Further KEGG pathway enrichment analysis revealed that prognostic DEGs

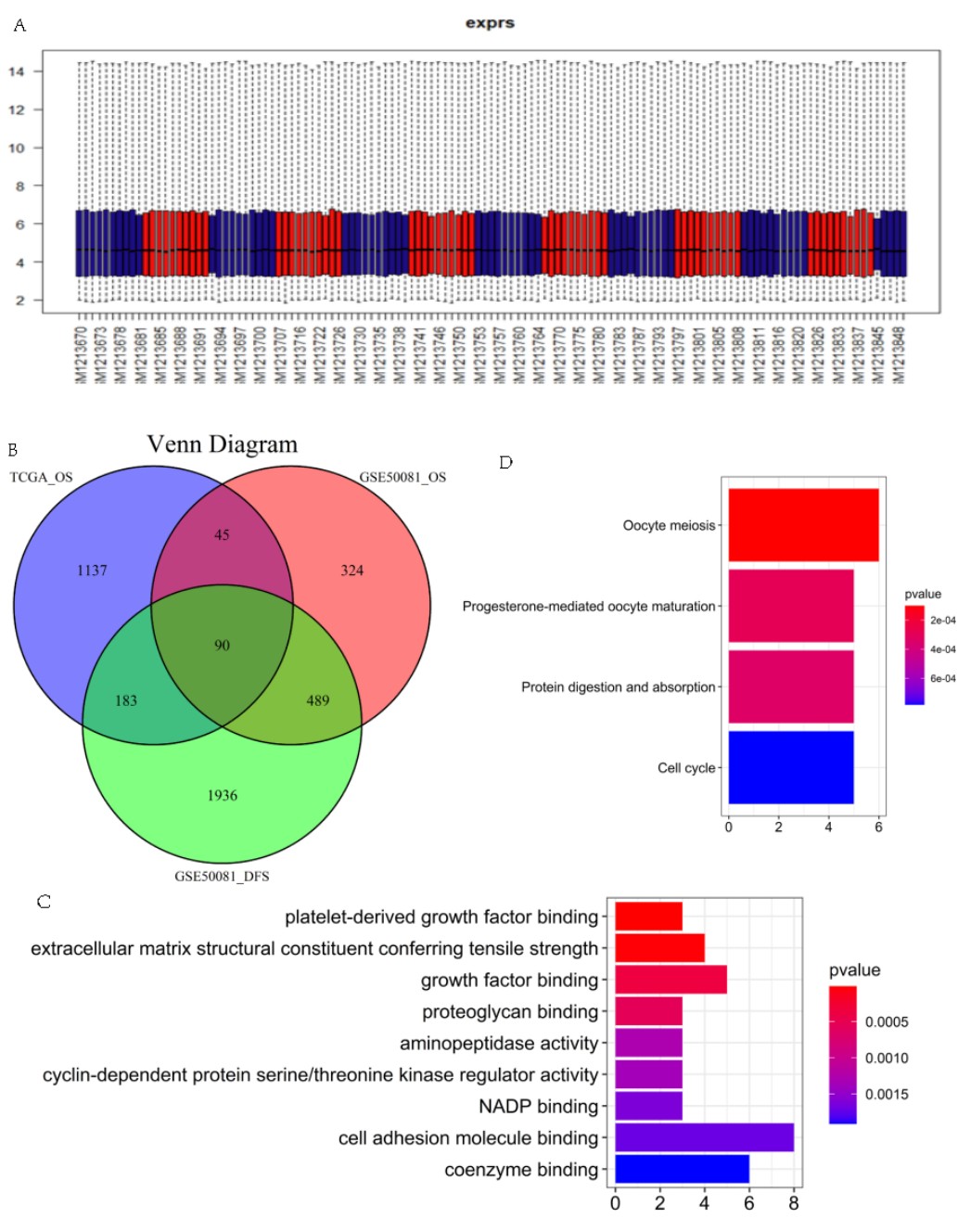

**Figure 2** (A) RMA normalization of the GSE50081 dataset. (B) Intersection of TCGA datasets, GSE50081 survival and relapse datasets. GO (C) and KEGG (D) pathway enrichment analysis of 90 prognostic genes from the TCGA dataset and the GSE50081 dataset.

were significantly enriched in "cell cycle", " protein digestion and absorption", "oocyte meiosis", and "progesterone-mediated oocyte maturation" pathways (Fig. 2D).

## Identification and verification of relapse-related DEGs

The most popular and effective treatment for early LUAD is radical surgical resection. Postoperative recurrence is one of the major risks of cancer-related death in early lung adenocarcinoma and also a significant cause of poor prognosis (*Martini et al., 1995*; *Mahvi et al., 2018*). In order to screen out more important target genes, we included the 90 prognostic genes we obtained into the GSE30219 prognosis and relapse dataset. Thirty-six genes showed significant differences (Table S4) and were subsequently included in the GSE30219 prognosis (Table S5) and recurrence dataset (Table S6) for LASSO regression analysis. To ensure result repeatability, 1,000 LASSO regressions were performed in each dataset. Genes that showed up more than 950 times in 1,000 analyses were considered significant. Ultimately, four genes were identified and subsequently used to construct a relapse-associated prognostic gene signature. The four genes identified were RAS protein activator like 2 (RASAL2, ENSG00000075391.15), cell division cycle associated 3 (CDCA3, ENSG00000111665.10), Fos-related antigen 1 (FOSL1, ENSG00000175592.7), and inositol polyphosphate-5-phosphatase B (INPP5B, ENSG00000204084.11). The risk score = $(0.8138293*$Exp RASAL2$) + (0.3749033*$Exp CDCA3$) + (0.3233154*$Exp FOSL1$) + (-1.3673670*$Exp INPP5B$)$. The Shapiro–Wilk test was used to calculate the distribution of risk scores in each dataset (Table S7). If data was normally distributed ($P > 0.05$) we selected their mean value as the cut-off. If the dataset had skewed distribution ($P < 0.05$), the median was selected as the cut-off. Samples were divided into high-and low-score subgroups according to their cut-off value. In order to verify the reliability of the risk score, we first conducted K-M analysis on three datasets, GSE30219, GSE50081, and TCGA respectively (Figs. 3A–3E). K-M curve analysis suggested that the high-score group predicted a poor prognosis compared with the low-score group, with statistically significant differences.

## Evaluating the independent role of the prognostic signature and construction of comprehensive prediction model

We conducted univariate regression analysis (Fig. 3F) for the clinical parameters of gender, age, smoking, T staging, and N staging (Table S8) of the GSE50081 dataset. We then performed multivariate cox regression analysis (Fig. 4A) for the results with a *P* value of <0.05 (T staging, N staging, and risk score). The results suggested that both univariate ($P = 0.006$, HR $=2.27$) and multivariate ($P = 0.014$, HR $= 2.10$) Cox regression analyses had significant prognostic significance in lymph node metastasis. Our analysis of clinical data from the TCGA dataset also found that only lymph node metastasis had significant prognostic value in both univariate and multivariate regression (Table S9). Therefore, we constructed a comprehensive survival prediction model (Fig. 4B) that included the two factors of lymph node metastasis and risk score.

We also calculated the correlation between clinical parameters and recurrence. The results indicated that there were significant statistical differences in the N stage for tumor recurrence regardless of the univariate ($P = 0.009$, HR $= 2.45$) (Fig. 4C) or multivariate regression analysis ($P = 0.037$, HR $= 2.08$) (Fig. 4D). T staging was statistically significant in univariate analysis ($P = 0.003$, HR $= 4.13$), but not in multivariate analysis ($P = 0.081$,

Peer

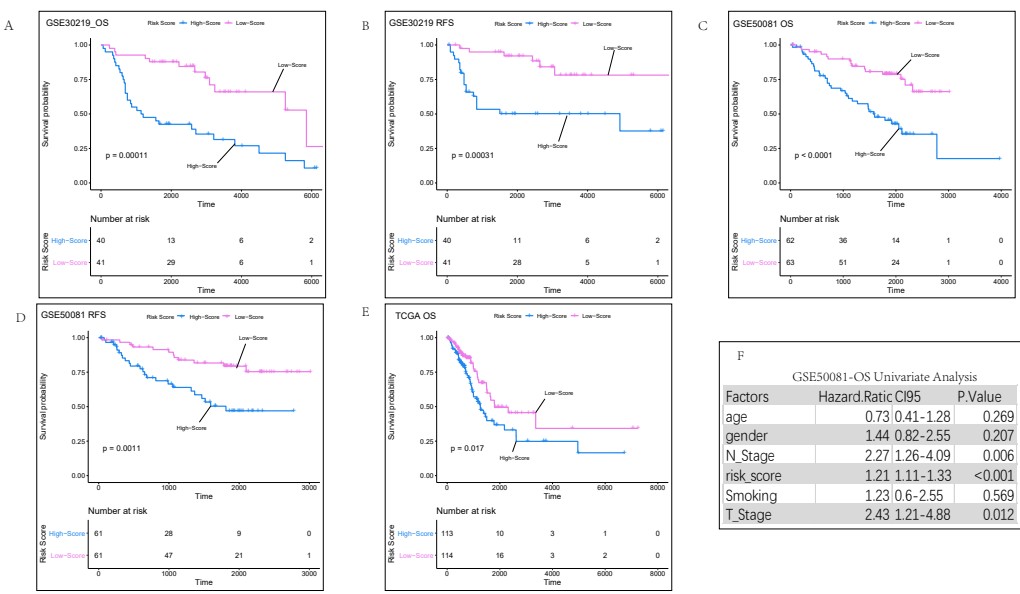

**Figure 3 KM analysis results of internal validation sets.** (A) K-M analysis of overall survival for risk scores in the GSE30219 dataset. (B) K-M analysis of relapse-free survival for risk scores in the GSE30219 dataset. (C) K-M analysis of overall survival for risk scores in the GSE50081 dataset. (B) K-M analysis of relapse-free survival for risk scores in the GSE50081 dataset. (E) K-M analysis of overall survival for risk scores in TCGA dataset. (F) Univariate regression analysis of clinical parameters related to overall survival in the GSE50081 dataset.

HR = 2.45). Therefore, we constructed a comprehensive recurrence prediction model (Fig. 4E) that included the lymph node metastasis and risk score.

## Internal validation of the prognostic and recurrent signature

To further evaluate the predictive performance and clinical suitability of our model, we evaluated the prognostic model and the recurrence model using time-dependent ROC curve analyses, calibration analysis and decision curve analysis (DCA), respectively. First, we verified the reliability of the survival prognosis model in the GSE50081 dataset, and ROC analysis found that the prediction ability of our model was generally good. Compared with the risk score univariate model (Fig. 5A), the combined model fitted with N stage and risk score (Fig. 5C) showed more advantages in the prediction of three-year survival. There was no significant difference in the prediction of five-year survival, while the combined model showed a relatively low ability in the prediction of one-year survival. This may be related to the poor prediction for one-year survival by the N staging univariate model (Fig. 5B). We performed a C-index analysis on the model to further investigate the clinical predictive power of the combined model. The C-index of the combined model was 0.694 (0.653–0.735), which indicates that the combined model has a good predictive ability. The c-index of the risk score univariate model was 0.683 (0.642–0.724). Further calibration analysis showed that our combined model had good overall clinical applicability. The combined model predicted three-year and five-year OS with higher accuracy than one-year OS (Figs. 5D–5F). We conducted a clinical DCA to evaluate the model's clinical efficacy

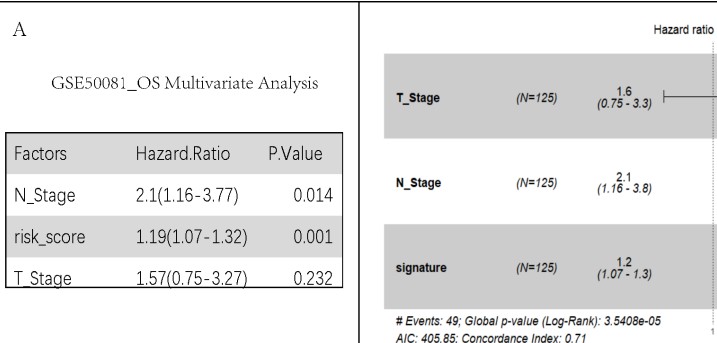

A

GSE50081_OS Multivariate Analysis

| Factors | Hazard.Ratio | P.Value |
|---------|--------------|---------|
| N_Stage | 2.1(1.16-3.77) | 0.014 |
| risk_score | 1.19(1.07-1.32) | 0.001 |
| T_Stage | 1.57(0.75-3.27) | 0.232 |

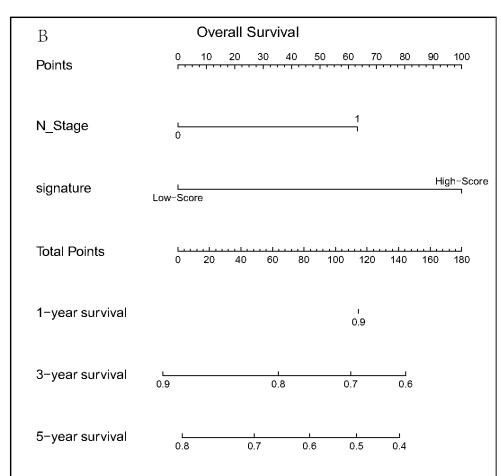

B — Overall Survival

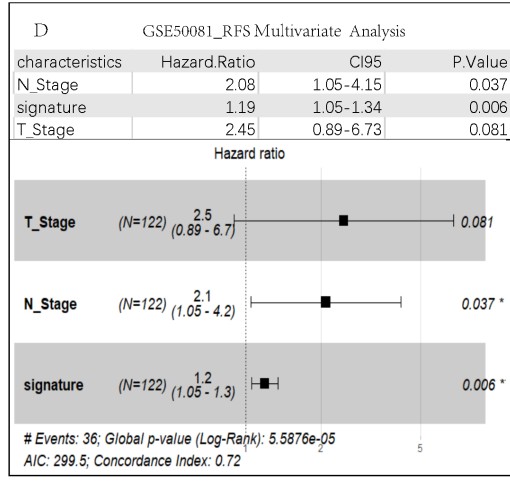

D

GSE50081_RFS Multivariate Analysis

| characteristics | Hazard.Ratio | CI95 | P.Value |
|-----------------|--------------|------|---------|
| N_Stage | 2.08 | 1.05-4.15 | 0.037 |
| signature | 1.19 | 1.05-1.34 | 0.006 |
| T_Stage | 2.45 | 0.89-6.73 | 0.081 |

C

GSE50081_RFS Univariate Analysis

| characteristics | Hazard.Ratio | CI95 | P.Value |
|-----------------|--------------|------|---------|
| gender | 1.29 | 0.67-2.5 | 0.444 |
| age | 1.22 | 0.63-2.36 | 0.547 |
| Smoking | 1.04 | 0.5-2.15 | 0.919 |
| T_Stage | 4.13 | 1.61-10.65 | 0.003 |
| N_Stage | 2.45 | 1.25-4.82 | 0.009 |
| risk_score | 1.23 | 1.1-1.37 | <0.001 |

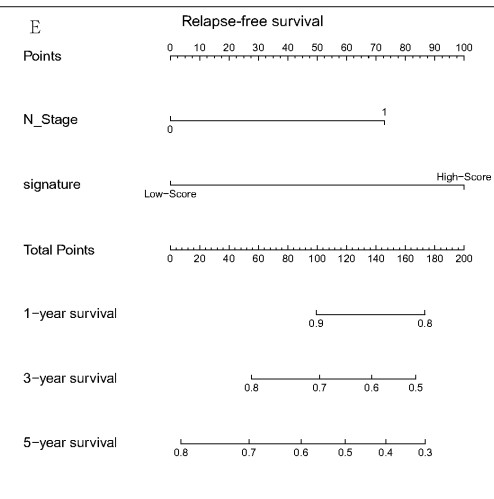

E — Relapse-free survival

**Figure 4 Construction of risk prediction model.** (A) Multivariate regression analysis of clinical parameters related to overall survival. For each patient, three lines are drawn upward to determine the points received from the three predictors in the nomogram. The sum of these points is located on the 'Total Points' axis. Then a line is drawn downward to determine the possibility of 1-, 3-, and 5-year OS (B)/RFS. (E) of LUAD. RFS-related univariate (C) and multivariate (D) regression analyses were performed to screen for independent relapse clinical parameters.

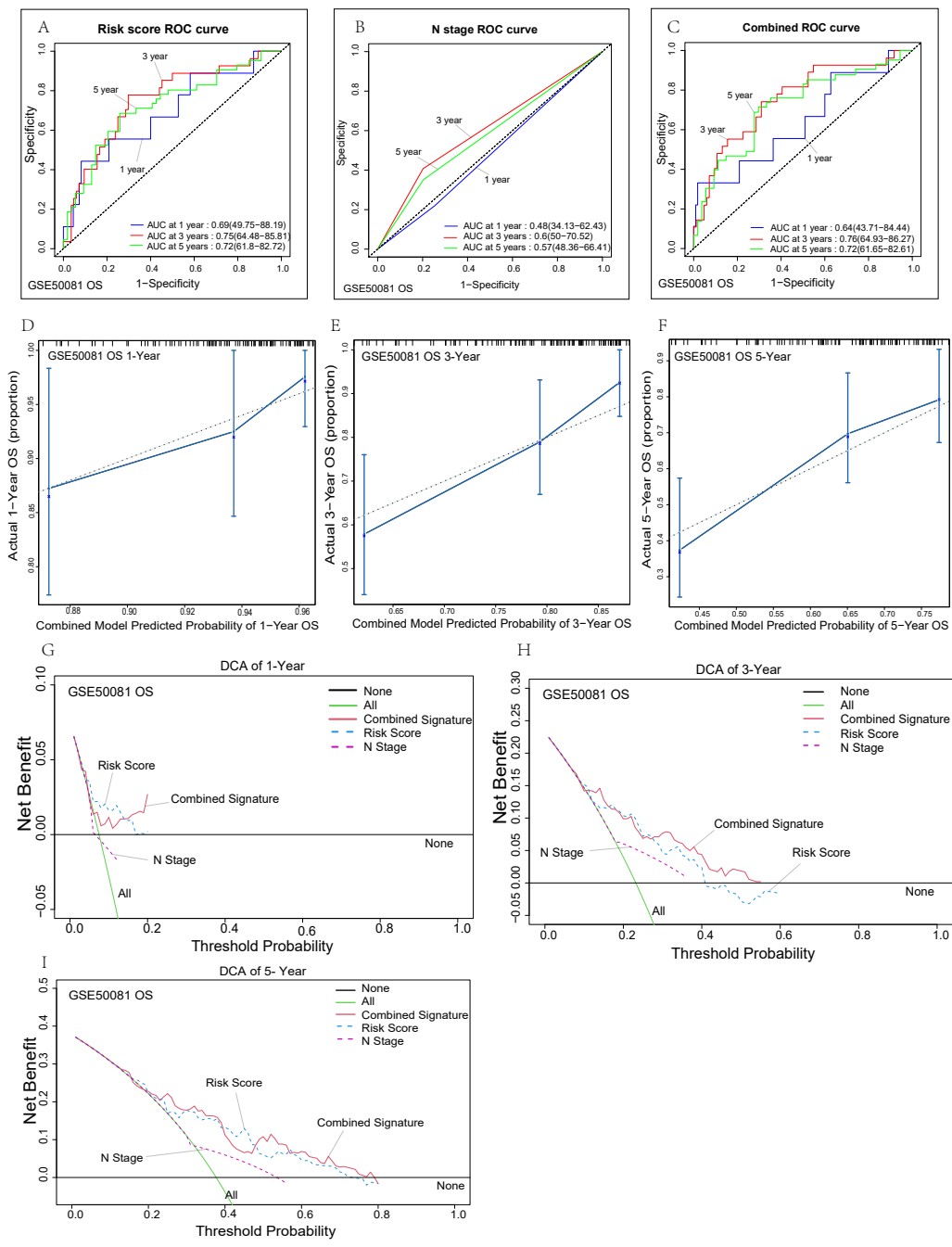

**Figure 5** **Verification of the risk signature.** (A–C) Time-dependent ROC curves of the nomogram of 1-, 3- and 5-year overall survival for univariate model and combined model to evaluate the predictive performance of the nomogram. The horizontal axis represents the sensitivity of nomogram prediction, and the vertical axis represents the specificity of nomogram prediction. The higher the sensitivity and specificity of nomogram curve, the stronger the prediction power. (D–F) The calibration plot for internal validation of the nomogram. The solid line represents the prediction ability of the model. The closer the solid line is to the dotted line, the closer the prediction of the model is to the real situation. (G-I) DCA curves of the univariate model and combined model for 1-,3- and 5-year overall survival in LUAD to evaluate the clinical decision-making benefits. The horizontal axis represents the threshold probability and the vertical axis represents the net gain. The higher the net gain under the same threshold probability, the stronger the Nomogram prediction power will be.

(Figs. 5G–5I). In general, our model had high predictive power and clinical consistency for three-year and five-year OS, while the one-year prediction model was relatively poor. The risk score model was similar to the combined model, while the lymph node prediction model had the worst predictive ability. In summary, through multiple algorithm validation, we found that our combined model has good long-term survival prediction ability, while the univariate risk score prediction model has good near-term prediction ability. This is reasonable because early lung adenocarcinoma has an extremely high one-year survival rate after surgery. Thus, we are more concerned about the long-term survival rate after surgery.

The reliability of the relapse-free survival prognostic model was verified in the GSE50081 dataset. ROC analysis showed that the combined model fitted with N stage and risk score had the best predictive power (Fig. 6A), with one-year, three-year, and five-year AUC of 0.78, 0.69, and 0.75, respectively. The risk score prediction model's predictive ability (Fig. 6B) was relatively high, while the predictive ability of the N-stage prediction model (Fig. 6C) was relatively poor. Lymph node metastasis is undoubtedly an independent prognostic factor for prognosis and recurrence, and our risk score model showed significantly better predictive power than the N Stage prognostic model, which will be more clinically applicable. We performed a C-index analysis on the model to further investigate the clinical predictive power of the combined model. The C-index of the combined model was 0.697 (0.647–0.747), which indicates that the combined model has good predictive ability. Further calibration analysis showed that our combined model had good clinical consistency in one-year, three-year, and five-year RFS (Figs. 6D–6F). Among them, our model has the highest prediction accuracy for one-year RFS, which is consistent with the above ROC analysis results. In addition, in order to evaluate the accuracy of the model and compare the advantages and disadvantages of each model, we conducted a clinical DCA (Figs. 6G–6I). In general, our model has high predictive power for one-year and five-year RFS. However, the combined model and the risk score model have their own advantages and disadvantages, and both are better than the N stage model. We found that our combined model has good RFS prediction ability through multiple algorithm validation. Similarly, we also conducted similar verification in the GSE30219 data set (Fig. S1). Due to the small sample size ($n = 2/81$) of lymph node metastasis in this dataset, we only validated the risk score model. The C-index of the survival and relapse groups was 0.758 (0.72–0.796) and 0.809 (0.765–0.853), respectively. In addition, the areas under the ROC curve in the survival group (Fig. S1A) and the relapse group (Fig. S1E) were 0.81, 0.83, 0.85, 0.88, 0.9, and 0.86 at one, three, and five years, respectively. These results show that the risk score prediction model has nearly perfect predictive ability.

## External validation of OS and RFS prognostic signature

The GSE31210 datasets were selected for external validation due to its complete prognosis and relapse sample information. The normalized results from the raw data are shown in Fig. S2. For the validation of survival related risk signatures, we adopted the same verification method described above. In the GSE31210 survival dataset ($n = 226$), we classified the risk scores into high-score and low-score groups. K-M analysis suggested

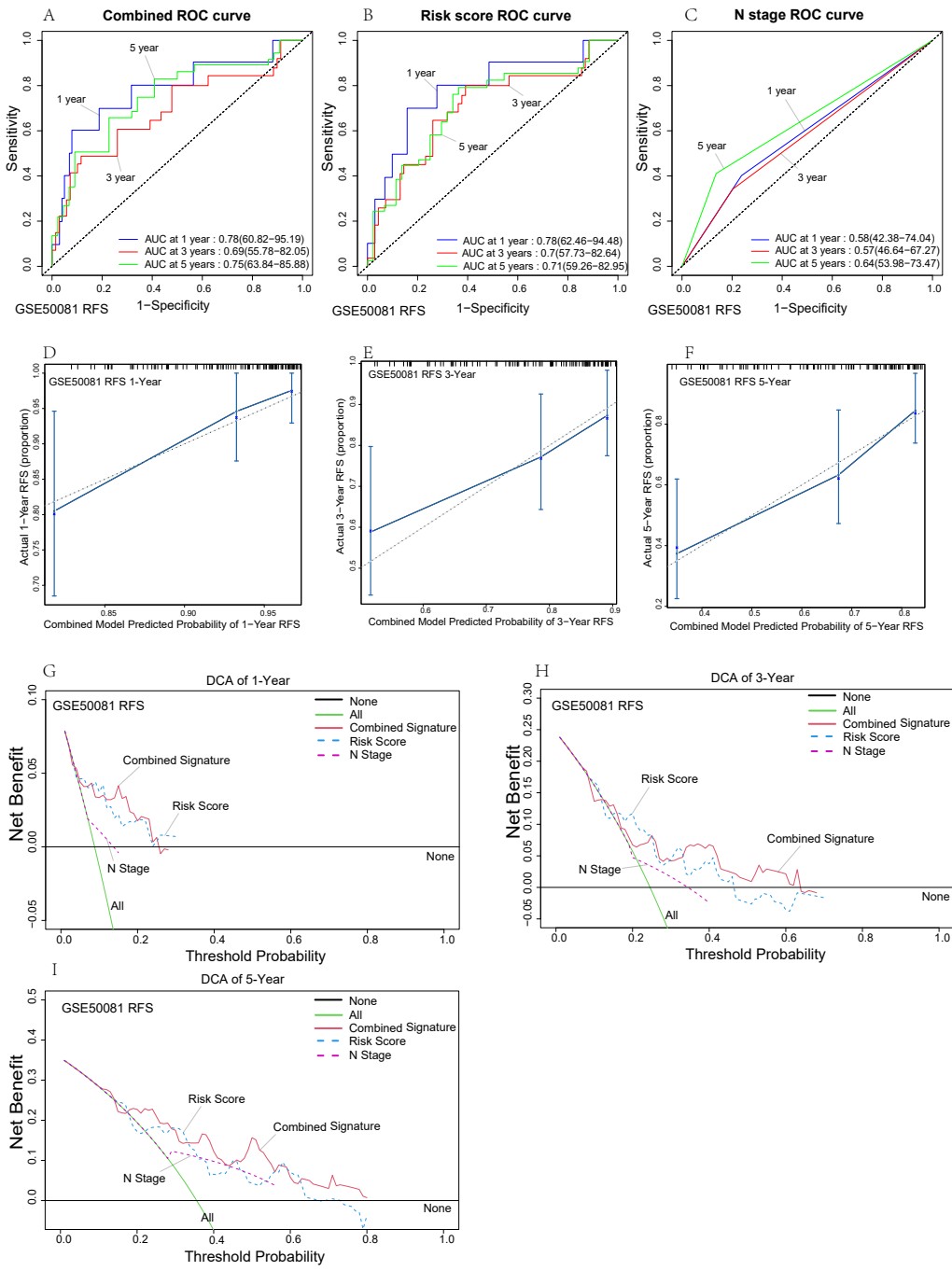

**Figure 6** **Internal validation of risk prediction models.** (A–C) Time-dependent ROC curves of the nomogram of 1-, 3- and 5-year RFS for univariate model and combined model to evaluate the predictive performance. (D–F) The calibration plot for internal validation of the nomogram. (G–I) DCA curves of the univariate model and combined model for 1-,3- and 5-year RFS in LUAD to evaluate the clinical decision-making benefts.

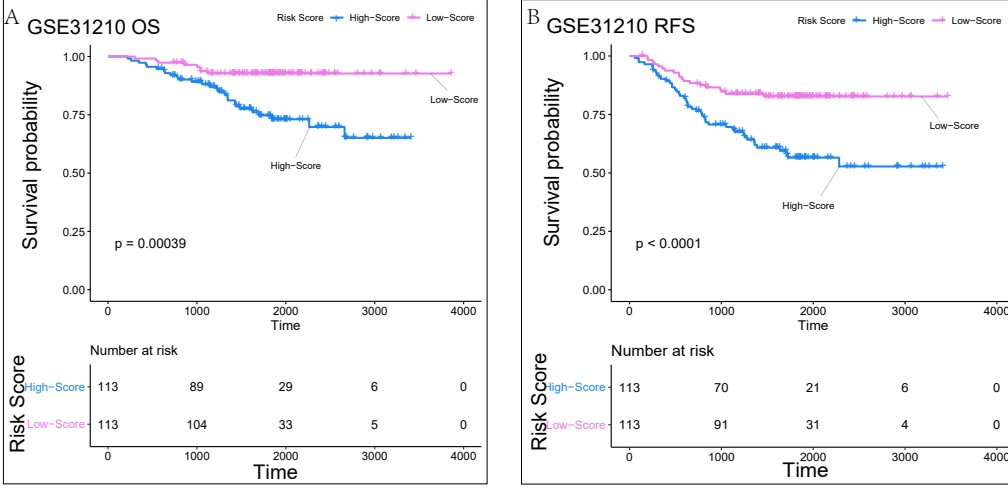

**Figure 7** **The KM analysis result of Risk Signature in the external verification set of GSE31210.** (A) The KM analysis result of Risk Signature on the overall lifetime. (B) The KM analysis results of Risk Signature for relapse-free survival.

that the high-score group had worse prognoses than the low-score groups ($P = 0.00039$) (Fig. 7A). Similarly, we found that the high-score group was more likely to relapse than the low-score group ($P < 0.0001$) (Fig. 7B). Time-dependent ROC analysis in the survival dataset showed that AUC for one-year, three-year and five-year OS of this validation set was 0.67, 0.66, 0.7, respectively (Fig. 8A). The C-index was 0.669 (0.628–0.71). Calibration analysis showed that the risk signature had good clinical consistency and survival prediction ability (Figs. 8B–8D). In the GSE31210 relapse dataset ($n = 226$), time-dependent ROC analysis showed that AUC for one-year, three-year, and five-year OS of this validation set was 0.65, 0.63, and 0.7, respectively (Fig. 8E). The C-index was 0.642 (0.609–0.675). Calibration analysis also showed good survival prediction (Figs. 8F–8H), especially for the five-year survival rate. In general, our risk signature has good predictive ability. Multiple algorithms show that our risk signature has a relatively good predictive ability in the GSE30219 dataset, which may be because the proportion of positive events in the GSE30219 dataset (42/81) is higher than that in the GSE31210 dataset (34/226).

## DISCUSSION

The clinical diagnosis of lung cancer has improved with the wide application of low-dose chest CT scan in routine screening of lung disease. Although minimally invasive radical surgery can greatly improve the survival rate of patients with cancer and has been widely used, there are still some patients who are prone to recurrence, which significantly affects their prognosis. Therefore, we constructed a 4-gene relapse-related survival risk model, which can identify individuals with a high recurrence rate and significant poor prognosis. Internal validation and external validation also confirm that our prediction model has high predictive ability. We can further develop individualized postoperative consolidation

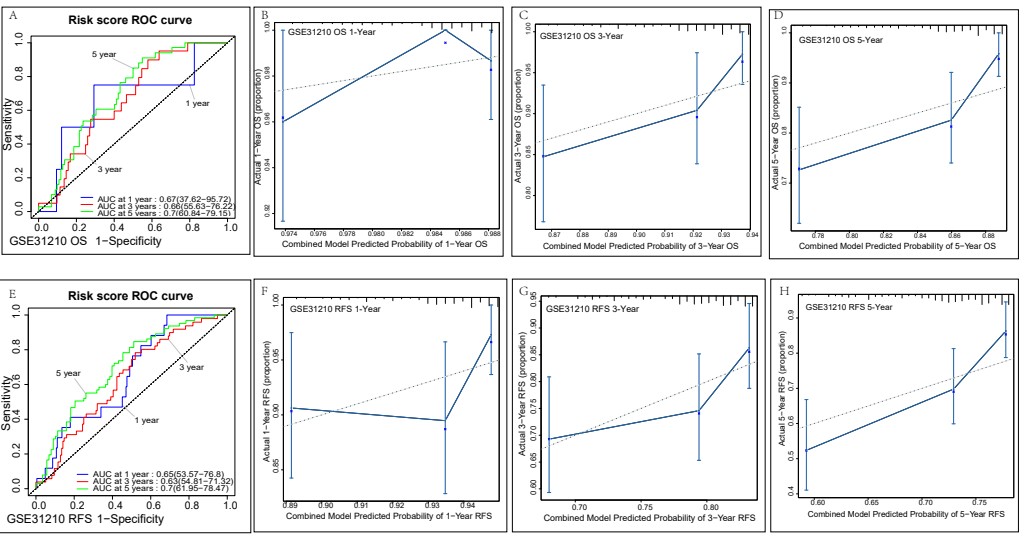

**Figure 8 External validation of genetic signatures.** (A)/(D)Time-dependent ROC curves of the risk signature for 1-, 3- and 5-year OS/RFS in LUAD to evaluate the predictive performance in the GSE31210 dataset. The calibration plot for 1- , 3- and 5-year OS/RFS of the risk signature in the GSE31210 dataset (B–D)/(F–H).

therapy, which is expected to improve overall survival after early lung adenocarcinoma for identified high-risk individuals.

A total of four genes were identified and validated (INPP5B, FOSL2, CDCA3, RASAL2). INPP5B seems to play a major role in our risk signature. The gene encodes a member of the inositol polyphospho-5 phosphatase family. These enzymes are involved in key activities such as cell signal transduction, membrane transport and cytoskeleton activity by regulating inositol phosphate (*Bohdanowicz et al., 2012*). Current studies on INPP5b have focused on its role in Lowe Syndrome and Type 2 DENT disease, while its role in lung cancer has not been reported. However, it has been shown that the same family of INPP5J appears to be a tumor suppressor gene (*Ben-Chetrit et al., 2015*), and many inositol polyphospho-5 phosphatase family molecules can affect cell migration, adhesion and polarity (*Ramos, Elong Edimo & Erneux, 2018*). For example, INPP5D (*Wain, Westwick & Ward, 2005*) and INPPL1 (*Venkatareddy et al., 2011*) have been reported to significantly affect cell migration and T cell chemotaxis. In addition, INPP5B is also significantly down-regulated in lung adenocarcinoma. Its high expression indicates a good prognosis and it may be a potential tumor suppressor gene in lung adenocarcinoma (http://gepia.cancer-pku.cn/detail.php?gene=INPP5B).

FOSL1 is a member of the Fos protein family. The protein encoded by FOSL1 is an essential component of the transcription factor complex AP-1 and is considered to be an important part of cell proliferation and differentiation (*Talotta, Casalino & Verde, 2020*). In lung cancer, FOSL1 acts as a bridge between KRAS mutations and mitosis and its inhibition will reduce the viability of KRAS mutant cells (*Vallejo et al., 2017*; *Elangovan et al., 2018*). FOSL1 can regulate gene expression induced by KRAS mutation, and thus

control cell proliferation and survival, and predicts a poor prognosis in lung cancer. In addition, FOSL1, as an important hub gene, can also be regulated by non-coding RNAs such as miR-130a and LINC00460 to further regulate the progression of lung cancer (*Xu, Wang & Liu, 2020*; *Cisneros-Villanueva et al., 2021*). Moreover, FOSL1 may play an important role in the development and metastasis of tumors (*Maurus et al., 2017*). It has been clearly reported that FOSL1 can also promote the progression of pancreatic cancer (*Vallejo et al., 2017*; *Luo, He & Qiu, 2018*), bile duct cancer (*Vallejo et al., 2021*), breast cancer (*Kim et al., 2020*; *Chen et al., 2018*), bladder cancer (*Cui et al., 2020*; *Gatta et al., 2019*), stomach cancer (*He et al., 2015*) and esophageal cancer (*Shen et al., 2020*) through direct or indirect mechanisms. Therefore, high expression of FOSL1 may contribute to poor prognosis of LUAD from multiple aspects of tumor genesis and progression.

CDCA3 is a member of the cell division cycle-associated protein family, which is involved in the cell cycle and has been found to promote cell proliferation and cell cycle processes in a variety of tumors (*Qian et al., 2018*; *Zhang et al., 2019b*; *Wu et al., 2020*). For example, in colorectal cancer, CDCA3 can mediate p21-dependent proliferation by regulating E2F1 expression or promote cell proliferation by activating the NF-KappaB /cyclin D1 signaling pathway (*Qian et al., 2018*; *Zhang et al., 2018*). In addition to promoting cell proliferation, CDCA3 has been reported to promote the migration, invasion and chemotherapy resistance of tumors (*Liu et al., 2020*; *Yu et al., 2020*). As an important hinge molecule, CDCA3 can also be directly regulated by non-coding RNAs (*Gao & Ji, 2021*; *Dou et al., 2020*; *Chen et al., 2020a*). These evidences suggest an important role of CDCA3 in promoting cancer in a variety of tumors. What's more, direct evidence for CDCA3 in lung cancer has also been provided. It has been reported that CDCA3 is significantly upregulated in lung cancer, the deletion of CDCA3 gene inhibits the proliferation of lung adenocarcinoma cell lines and promotes cell senescence (*Adams et al., 2017*), which is consistent with our results.

RASAL2 is a member of the RAS GTPASe-activated proteins (GAP) family, which negatively regulate the RAS signaling pathway by catalyzing the hydrolysis of Ras-GTP to Ras-GDP (*Zhou et al., 2019*). Several studies have shown that its effect on tumors is bidirectional (*Zhou et al., 2019*; *Wang, Yin & Yang, 2019*; *Wang et al., 2019*). For instance, RASAL2 inhibits tumor cell migration and invasion by inactivating the RAS pathway (*Hui et al., 2017*; *McLaughlin et al., 2013*). RASAL2 can also promote the progression of colorectal cancer through the LATS2 / YAP axis (*Pan et al., 2018*). In lung cancer, previous studies seem to indicate that RASAL2 is a tumor suppressor (*Xiong et al., 2021*). Low expression of RASAL2 can promote EMT in lung cancer (*Li & Li, 2014*) and promote metastasis through the RAS /ERK pathway (*Fan et al., 2021*), which is inconsistent with our results. Our evidence suggests that RASAL2 is a contributing factor to poor prognosis in lung adenocarcinoma. This is an interesting conclusion, and paradoxical conclusions must be accompanied by important intermediate mechanisms. A recent study suggests that the phosphorylation status of RASAl2 S351 acts as a molecular switch to inhibit or promote AMPK-mediated autophagy (*Bao et al., 2021*), suggesting that there is a regulatory switch for the bidirectional effect of RASAL2.

In conclusion, INPP5B is a potential tumor suppressor gene that has not yet been experimented, and RASAL2 may have a bidirectional effect on tumors, however, both

|  | INPP5B | FOSL1 | CDCA3 | RASAL2 |
|---|---|---|---|---|
| Other Cancers | – | ◯ | ◯ | ◗ |
| Lung cancer | – | ◯ | ◯ | ● |
| Our results | ● | ◯ | ◯ | ◯ |

– Unreported  ● Antitumor effect

◯ Tumor-promoting effect

◗ Bidirectional effect

**Figure 9** **The effect of key genes reported from previous studies compared with our analysis.**

require further study. CDCA3 and FOSL1 have relatively clear carcinogenic effects and are indicators of poor prognosis (Fig. 9) (*Chen et al., 2020b*). Our risk signature had a good ability to predict both postoperative RFS and OS for early lung adenocarcinoma. However, there are still some shortcomings in our analysis. First, the combined model with clinical parameters was not validated in the external dataset, mainly because some clinical data in the external dataset were incomplete or severely unbalanced. Second, we require additional external data sets containing complete information to further verify our model's predictive ability. Third, regarding the cut-off value of the risk signature, due to the reagent, sequencing time/method, operational differences, and other factors, the batch effect between different datasets was too large. There is still an irreconcilable batch effect between different datasets after the SVA package was used. A uniform cut-off value, therefore, is not suitable for this study. Fourth, as a predictive model, the combined model may underestimate or overestimate patient outcomes.

## CONCLUSIONS

We constructed a four-gene signature risk model to predict RFS and OS after early lung adenocarcinoma surgery. We found that both CDCA3 and FOL1 have clear tumor-promoting effects in a variety of tumors and both play key roles in the regulatory mechanisms, which is consistent with previous studies. We also found that INPP5B may be a potential tumor suppressor gene that has never been reported in tumors, and there is a variety of evidence to demonstrate the important role of inositol polyphospho-5 phosphatase family in signal transduction, cell adhesion, and migration. Interestingly,

RASAL2 shows significant bidirectional effects in other tumors, and it has been reported that the epigenetic regulation of RasAL2 acts as a molecular switch for promoting and inhibiting effects. Therefore, the complex mechanism of action of RASAL2 requires further clarification. In conclusion, our study may improve our understanding of the postoperative progress of early lung adenocarcinoma and provide new diagnostic indicators and therapeutic targets for clinical treatment in the future.

### Funding

The authors received no funding for this work.

### Competing Interests

The authors declare there are no competing interests.

### Author Contributions

- Peng Han conceived and designed the experiments, analyzed the data, prepared figures and/or tables, authored or reviewed drafts of the paper, and approved the final draft.
- Jiaqi Yue analyzed the data, prepared figures and/or tables, authored or reviewed drafts of the paper, and approved the final draft.
- Kangle Kong, Shan Hu and Peng Cao performed the experiments, prepared figures and/or tables, and approved the final draft.
- Yu Deng conceived and designed the experiments, performed the experiments, prepared figures and/or tables, and approved the final draft.
- Fan Li conceived and designed the experiments, analyzed the data, prepared figures and/or tables, and approved the final draft.
- Bo Zhao conceived and designed the experiments, prepared figures and/or tables, and approved the final draft.

### Data Availability

The R code and related files are available in the Supplemental Files.

### Supplemental Information

Supplemental information for this article can be found online at http://dx.doi.org/10.7717/peerj.11923#supplemental-information.

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
