# Peer review of "Signature identification of relapse-related overall survival of early lung adenocarcinoma after radical surgery"

_PeerJ, doi:10.7717/peerj.11923_

## Round 0.1 · original submission · Major Revisions

· Academic Editor

Major Revisions

In addition to the constructive and objective comments from the two reviewers, I have the following concerns regarding your manuscript:

1. Some pictures are not clear and their high-resolution replacements are required.

2. There are mistakes and grammar errors throughout the whole manuscript, such as the repeated description of GO and KEGG in Lines 88-93 and the wrong citation of Sub-Figures in lines 239-240 etc.

When revising your manuscript, please consider all issues mentioned in our comments CAREFULLY and SERIOUSLY, and provide suitable responses for any comment. Please note that your revised submission may need to be re-reviewed.

PeerJ values your contribution and I look forward to receiving your revised manuscript.

Reviewer 1 ·

Basic reporting

Overall, the manuscript writing is clear and easy to understand, except some examples where the language could be improved for better comprehension, e.g. line 71-72, “We found that some patients…or patients with multiple…”; line 75-76, “on the other hand…after the first operation”.

Experimental design

I have a question on the selected TCGA RNA-seq dataset for differential expression (DE) analysis. Are the LUAD patients developed recurrent LUAD after their 1st operation? If not, why the DEGs between LUAD vs. normal lung tissue could be used for identifying prognostic signature? My understanding is, the DEGs should come from comparing the post-operative LUAD patients who showed recurrence vs. the post-operative LUAD patients who did not. Although, if the authors could provide more evidence that the LUAD vs. normal DEGs have direct correlation with prognostic recurrence, it’d be more logical to doing DE analysis between LUAD and normal lung tissues. A side note: I do notice that the authors later used a more relapse-related GSE5081 dataset for further screening. This dataset fits better to the signature identification target.

Validity of the findings

w.r.t external validation, is the high/low risk cut-off value selected in the validation set or the training set? Most common practice for biomarker identification is, these types of hyper parameters were determined in the training set for independent prediction model validation. If there is batch effect between the training set and validation set, I would recommend the authors to consider using the R package sva and removing the batch effect before model validation.

Additional comments

In the linear function of risk score, gene “PRR11” has >2x weight than the other three genes. I’m curious if the authors have dived deeper to explain why PRR11 is the driver gene among the 4 genes?

·

Basic reporting

Overall, the design and analysis of the study is sound, with clear results and relatively straightforward conclusions. Regarding reporting, I recommend the following to improve readability of the manuscripts:

Line 62: the sentence "accelerated the pace of national physical examination" is bit unclear. It if is referring to a specific nation, then it should state it more explicitly. Recommend rephrasing.

LIne71- Line80 this seems like a very lengthy discussion about second operation, which isn't directly relevant to the focus of the paper. Recommend streamlining.

Line84 - Line96 Similarly, this part seems like a very lengthy introduction of the methodology. This part could be streamlined or details moved to methods section.

Line197: Description of the text does not match the figure. The text is referring Fig2E.

Line202: Description of the text does not match the figure. The text is referring to Fig2D. Fig2D and Fig2E should be switched according to the text.

Line214: Fig3A-C y axis title can be modified to reflect overall survival, relapse-free survival and overall survival respectively.

Line218-219 Readers could benefit from having more context on what GSE50081 study has already done.

Line228: Fig3D and Fig4B should each be labelled with a title to allow reader to more easily distinguish the difference. One is for overall survival, the other one is for recurrence.

Line248: the axis ticks, labels are all very small and hard to read. Recommend enlarging text size.

Line248-249: It would be very helpful if the authors can guide readers on these figures, and provide more descriptive details on how they interpreted Figures 5D-5I before reaching conclusions.

Most of the figure legend main titles could benefit from being more descriptive. Recommend adding key information and takeaways for figure legend main titles. Especially for Fig3-Fig8.

Experimental design

Line175: In general, the software/packages versions should also be reported for reproducibility. Which R version is used?

Line176: Is the p value referred here after correction for multiple testing? If so what method (FDR, BH etc.)

Line187: the font size of the pathway text are very small and hard to read. Recommend increasing text font size.

Line208 and Line280: In the supplementary text1, the authors showed the normality test and p values, however it would improve clarity if they can also show how they determined risk score cut-off values based on normality test.

Line203-line204: Could the authors explain how the four genes were screened out from the K-M analysis. How and why were these four genes chosen for downstream signature analysis? There seems a missing methodology step here.

Validity of the findings

Overall well done study from the authors. The data analysis and results justifies the conclusions.

Additional comments

The prognostic and recurrence biomarkers has great values, and I believe this study will be of interest to scientists and clinicians involved in lung adenocarcinoma research as well as cancer biology in general. There are some areas the authors could modify to improve readability and clarity, as outlined above.

---

## Round 0.2 · Minor Revisions

· Academic Editor

Minor Revisions

Thank you for your revision. I agreed with the reviewer 2 on your much improved manuscript. However, I think that you may need to perform a complete literature search and to discuss more about four hub genes (INPP5B, FOSL2, CDCA3, RASAL2). Currently, they are not enough. For examples, I did not fully agree with you on "The role of Rasal2 in lung cancer remains to be studied". At least there exist two articles to study the role of Rasal2 in lung cancer and you did not mention it:

1. miR‑654‑3p suppresses cell viability and promotes apoptosis by targeting RASAL2 in non‑small‑cell lung cancer, Mol Med Rep
. 2021 Feb;23(2):124

2. RASAL2 promotes lung cancer metastasis through epithelial-mesenchymal transition, Biochem Biophys Res Commun
. 2014 Dec 12;455(3-4):358-62.

I would also prefer you to read the article (Identification of Prognostic miRNA Signature and Lymph Node Metastasis-Related Key Genes in Cervical Cancer, Front Pharmacol. 2020 May 8;11:544. This is my article, you do not need to cite this article if you did not imitate to draw a Table similar to Table 4 in my article ) for discussing the hub genes.

A detailed discussion can help the readers to recognize the reliability and predictability of these key genes which you identified using bioinformatic analysis.

By the way, it would be better for you to provide the date of data downloaded.

Finally, please carefully check the manuscript and the supplementary files before you submit the new version.

·

Basic reporting

Much improved readibility and flow of paper.

Experimental design

The updated and optimized study design has made the study more sound and rigorous.

Validity of the findings

no comment.

Additional comments

Great work and much improved compared to the original manuscript after editing. I recommend it for publication.

---

## Round 0.3 · accepted · Accept

· Academic Editor

Accept

You need to revise two minor things during the proofreading.

1) You need to add or label “RASAL2: ENSG00000075391.15; CDCA3: ENSG00000111665.10; FOSL1: ENSG00000175592.7; INPP5B: ENSG00000204084.11” and mention their abnormal up/down-regulations in the result parts (maybe in lines 170-173, also noted with supplementary Table 1); otherwise the up-regulations of RASAL2, CDCA3, FOSL1 and the down-regulation of INPP5B in lung cancer you mentioned in the discussion part appear abruptly.

2) That “We found that both CDCA3 and FOL1 have clear tumor-promoting effects in a variety of tumors and both play key roles in the regulatory mechanisms, which is consistent with our conclusion.” is obviously not right. It should be "consistent with previous studies" or something else.